# SMAD4 Overexpression in Patients with Sleep Apnoea May Be Associated with Cardiometabolic Comorbidities

**DOI:** 10.3390/jcm9082378

**Published:** 2020-07-25

**Authors:** Elena Díaz-García, Ana Jaureguizar, Raquel Casitas, Sara García-Tovar, Begoña Sánchez-Sánchez, Ester Zamarrón, Eduardo López-Collazo, Francisco García-Río, Carolina Cubillos-Zapata

**Affiliations:** 1Biomedical Research Networking Centre on Respiratory Diseases (CIBERES), 28029 Madrid, Spain; elena_dg_1994@hotmail.com (E.D.-G.); anajaureguizaroriol@gmail.com (A.J.); rqlkas@yahoo.es (R.C.); carolinacubillos@hotmail.com (B.S.-S.); ester.zamarron@gmail.com (E.Z.); elcollazo@hotmail.com (E.L.-C.); fgr01m@gmail.com (F.G.-R.); 2Respiratory Diseases Group, Respiratory Diseases Department, La Paz University Hospital, IdiPAZ, 28046 Madrid, Spain; sarugarto@gmail.com; 3The Innate Immune Response Group, La Paz University Hospital, IdiPAZ, 28046 Madrid, Spain; 4Faculty of Medicine, Autonomous University of Madrid, 28029 Madrid, Spain

**Keywords:** OSA, TGFβ, SMAD4, HIF1α, intermittent hypoxia, circadian rhythm

## Abstract

Obstructive sleep apnoea (OSA) is associated with several diseases related to metabolic and cardiovascular risk. Although the mechanisms involved in the development of these disorders may vary, OSA patients frequently present an increase in transforming growth factor beta (TGFβ), the activity of which is higher still in patients with hypertension, diabetes or cardiovascular morbidity. Smad4 is a member of the small mother against decapentaplegic homologue (Smad) family of signal transducers and acts as a central mediator of TGFβ signalling pathways. In this study, we evaluate Smad4 protein and mRNA expression from 52 newly diagnosed OSA patients, with an apnoea–hypopnoea index (AHI) ≥30 and 26 healthy volunteers. These analyses reveal that OSA patients exhibit high levels of SMAD4 which correlates with variation in HIF1α, mTOR and circadian genes. Moreover, we associated high concentrations of Smad4 plasma protein with the presence of diabetes, dyslipidaemia and hypertension in these patients. Results suggest that increased levels of SMAD4, mediated by intermittent hypoxaemia and circadian rhythm deregulation, may be associated with cardiometabolic comorbidities in patients with sleep apnoea.

## 1. Introduction

Obstructive sleep apnoea (OSA) is a very prevalent disorder which affects around 936 million adults worldwide [1], characterized by recurrent episodes of partial or complete upper airway obstruction associated with intermittent hypoxia and sleep fragmentation. These alterations induce oxidative stress, systemic inflammation, sympathetic activation, and metabolic deregulation [2], and are associated with a high risk of cardiovascular and metabolic disease [3,4,5].

Although the mechanisms involved in the development of these disorders may vary, OSA patients frequently present an increase in transforming growth factor beta (TGFβ) levels [6], suggesting that the pathways dependent on its activation could play a central role. TGFβ belongs to a signalling super family which has multiple biological functions, including roles in development, immune response, and cell growth [7]. High plasma levels of TGFβ have been reported in patients with hypertension, type 2 diabetes and cardiovascular diseases, suggesting a possible role in their development [8,9]. Therefore, for example, the pleiotropic effects of TGFβ signalling on metabolism and energy homeostasis are relevant to the aetiology and progression of diabetes [7].

Smad (small mother against decapentaplegic homologue) proteins mediate TGFβ signalling [10]. Briefly, TGFβ receptors phosphorylate receptor (R-) Smads (Smad2 and Smad3) [10,11]; then phosphorylated R-Smads oligomerize with the common mediator (Co-) Smad4; finally, this SMAD4 complex translocates to the nucleus and regulates gene expression. Therefore, Smad4 is highly regulated by ubiquitination [11,12] and deubiquitination [13] in order to control the SMAD complex formation. In addition, Smad4 has a multiface regulator effect on metabolic homeostasis, promoting fibrosis, fibroblast and matrix remodelling, and inflammation [14]. These processes highlight the importance of Smad4.

Both intermittent hypoxia and sleep fragmentation due to OSA may have some impact on the regulation of the TFGβ–Smad pathway. As a consequence of sleep fragmentation, OSA patients suffer daytime sleepiness, which leads to deregulation of the circadian rhythm [15]. The molecular regulation of circadian rhythm is afforded by the core oscillator, brain and muscle arnt-like 1 (BMAL1), which can heterodimerize with circadian locomotor output cycles kaput (CLOCK) or neuronal PAS domain-containing protein 2 (NPAS2); both CLOCK:BMAL or NPAS2:BMAL heterodimers bind to the E-box-enhancer elements, activating the transcription of a negative feedback loop composed by Period (PER1-3) and Cryptochrome (CRY1-2) genes [16,17]. Growing evidence has led to the elucidation of the physiological roles of the intrinsic circadian clock and maintenance of tissue homeostasis, cellular metabolism and inflammatory processes [18,19,20].

Additionally, intermittent hypoxia (IH) triggers the hypoxia inducible factor (HIF1α) in OSA patients, in in vitro and in animal models [6,21,22]. Moreover, HIF1α expression induces TGFβ expression, as we have previously reported in OSA patients [6]. We have demonstrated that IH drives the immunosuppressive phenotype by the activation of TGFβ through OSA monocytes [23]. Besides, different experimental models have shown that the upregulation of TGF-β1 gene expression is a consequence of the acute sleep fragmentation [24,25,26]. Indeed, TGF-β1 has been demonstrated to display a circadian rhythm expression pattern in lung cells [26], chondrocytes [27] and fibroblasts [28].

As Smad4 is an important player in the regulation of the canonical pathway of TFGβ susceptible to intermittent hypoxia and circadian rhythm deregulation, we analyse its activity in OSA patients. Firstly, we analyse the relationship between indices of hypoxaemia or circadian rhythm and SMAD4 expression in OSA patients. Moreover, we explore the association between Smad4 expression and cardiometabolic comorbidities of OSA.

## 2. Experimental Section

### 2.1. Study Participants

Newly diagnosed OSA patients over 35 years of age with an apnoea–hypopnoea index (AHI) ≥30 on respiratory polygraphy were consecutively recruited from the sleep unit of La Paz University Hospital-Cantoblanco, Madrid, Spain. The diagnosis of OSA was established by respiratory polygraphy (Embletta GOLD, ResMed), which included a continuous recording of oronasal flow and pressure, heart rate, thoracic and abdominal respiratory movements and oxygen saturation (SpO2). Those tests in which the patients claimed to sleep less than 4 h or in which there were less than 5 h of nocturnal recording were repeated. Exclusion criteria were the following: previous or current treatment with oxygen or mechanical ventilation; underweight (body mass index (BMI) < 18.5 kg/m^2^) or morbid obesity (BMI > 40 kg/m^2^); history of respiratory disease, including chronic obstructive pulmonary disease, asthma or respiratory failure; any infectious disease in the previous 3 months; and the use of inhaled or systemic corticosteroids or other anti-inflammatory drugs.

For every two patients with severe OSA included in the case group, a control subject was randomly selected from the census register of the Madrid, Spain metropolitan area, who was paired for sex, age (±2 years) and body mass index (±5 kg/m^2^) with the second case. Selected subjects were considered as control when they had no known history of respiratory or cardiovascular disease or previous use of any type of medication. Moreover, the diagnosis of OSA was ruled out by respiratory polygraphy.

A summary of the distribution of the participants is detailed in Appendix A. The study was approved by the local ethics committee (PI-3646), and informed consent was obtained from all participants. 

Early in the morning following the sleep study, fasting blood samples were taken from all enrolled participants and blood pressure was measured in a supine position after at least 15 min of rest. Hypertension was defined as current usage of antihypertensive medication or if the mean of three blood pressure readings exceeded 140/90 mmHg. Dyslipidaemia was diagnosed by the presence of anamnesis and the use of lipid-lowering drugs or the presence of dyslipidaemic serum levels. For blood analysis, the following cut-off values were defined as abnormal: Total cholesterol ≥ 200 mg/dL, triglycerides ≥ 180 mg/dL, high-density lipoprotein (HDL)-cholesterol ≤ 40 mg/dl, and low-density lipoprotein (LDL)-cholesterol ≥ 150 mg/dL [29]. Type 2 diabetes was diagnosed when a patient fulfilled at least one of the following criteria: current treatment with oral anti-diabetic drugs and/or insulin, a fasting glucose level above 126 mg/dL on at least 2 separate occasions, glycated haemoglobin (HbA1c) level > 6.5%, or blood glucose level 2 h after oral glucose tolerance test equal to or more than 200 mg/dL.

### 2.2. PBMC Isolation and Culture

Peripheral blood mononuclear cells (PBMCs) from OSA patients and healthy volunteers were isolated by Ficoll-Paque Plus (Amersham Bioscience, Uppsala, Sweden) gradient by centrifugation. Cells were maintained in Roswell Park Memorial Institute (RPMI) 1640 medium supplemented with 100 U/mL penicillin and 100 μg/mL streptomycin and without foetal bovine serum (FBS) in an adherent surface treatment for 1 h at 37 °C and 5% CO_2_ to enrich the monocytes. We seeded 0.5 × 10^6^ monocytes per well (6-well plates). The medium was then replaced with fresh culture media supplemented with 10% fetal bovine serum. Cells were incubated at 37 °C and 5% CO_2_ overnight. 

### 2.3. Intermittent Hypoxia In Vitro Model 

Intermittent hypoxia (IH) models were performed using an incubator chamber attached to an external O_2_/N_2_ computer-driven controller using BioSpherix OxyCycler-42 (Redfield, NY, USA). This strategy mimicked the cyclical changes in O_2_ concentrations that control air gas levels, maintaining CO_2_ as previously described in detail [21]. Briefly, the IH model cycled O_2_ saturation in the medium at 1% for 2 min, followed by 20% for 10 min, with CO_2_ maintained at 5%. The total number of cycles was 72 during 16 h. Meanwhile, cells from each subject were also cultured under normoxic conditions (21% O_2_, 5% CO_2_, 37 °C) for the control group.

### 2.4. HIF1α Inhibition

HIF1α inhibition was performed using monocytes from HV subjects. Monocytes were treated with 15 μM of PX-478 from MedKoo Biosciences Inc. (Research Triangle Park, NC, USA) [30,31] overnight [31].

### 2.5. mRNA Isolation and Quantification

The monocytes cell culture was harvested and washed with phosphate-buffered saline. The RNA was isolated using the High Pure RNA Isolation Kit (Roche Diagnostics, Basel, Switzerland). Then, complementary DNA (cDNA) was obtained through reverse transcription using the High-Capacity cDNA Reverse Transcription kit (Applied Biosystems, Foster City, CA, USA). mRNA quantification was assessed by real-time quantitative PCR using LightCycler system (Roche Diagnotics, Basel, Switzerland) and QuantiMix Easy SYG kit from Biotools (Madrid, Spain) with specific primers. The results were normalized to the expression of 18S, and the cDNA copy number of each gene of interest was determined using a 6-point standard curve. All primers were synthesized, desalted, and purified by Eurofins Scientific SE (Luxembourg).

### 2.6. Determination of Plasma Levels of Soluble Proteins

The blood samples were centrifuged to separate plasma, and all specimens were immediately aliquoted, frozen and stored at −80 °C. Smad4 (CSB-E12749 from CUSABIO, Houston Texas, TX, USA) was assayed using the human enzyme-linked immunosorbent assay (ELISA). We followed the manufacturer’s instructions for all samples. Measurements for plasma samples were done in duplicate. The detection limit of the assay was 6.25 pg/mL. The intra-assay variation was below 20% in the two different assays.

### 2.7. Statistical Analyses

Statistical significance between healthy volunteers and OSA patients was calculated using an unpaired *t*-test with or without Welch’s correction according to data distribution. A two-way ANOVA analysis was used to compare differences between groups of data. In addition, correlations were assessed with Spearman’s correlation coefficient. To examine associations between Smad4 levels (per 50 pg/mL) and comorbidities, multiple logistic regression models with adjustment for age, sex and BMI were developed. The differences were considered significant at *p* < 0.05, and the analyses were conducted using Prism 8.0 software (GraphPad, San Diego, CA, USA).

## 3. Results

### 3.1. Study Subjects

Fifty-two patients with severe OSA and 26 healthy volunteers (HV) were included in this study. Their anthropometric and sleep characteristics are shown in Table 1. Thirty-five OSA patients presented some morbidity, the most frequent being dyslipidaemia (56%), hypertension (54%), diabetes (31%) and ischaemic heart disease (13%). The anthropometric characteristics, the apnoea-hypopnea index and the other sleep parameters were similar between OSA patients with or without comorbidity.

### 3.2. SMAD4 is Overexpressed in OSA Patients 

We found a higher expression of SMAD4 in OSA monocytes in comparison with HV monocytes (*p* = 0.0102). Similarly, the plasma levels of Smad4 were also higher in OSA patients than in HV (*p* < 0.0001) (Figure 1).

### 3.3. Hypoxaemia Enhances SMAD4 Expression

To assess if OSA-induced hypoxaemia triggers TGFβ/Smad4 activity, we firstly corroborated the two most contrasted hypoxia transcription factors, HIF1α and mTOR increase in OSA patients compared to HV. Then, we evaluated the correlation between SMAD4 expression with the HIF1α and mTOR expression. In OSA patients, SMAD4 mRNA expression was significantly associated both with HIF1α (*r* = 0.4100, *p* = 0.0095) and mTOR mRNA expression (*r* = 0.3304 and *p* = 0.0373) (Figure 2A,B).

In order to verify the role of HIF1α in this context, we used an in vitro model of IH with monocytes from healthy subjects as we have previously described [21]. Additionally, we treated HV monocytes with PX-478 (*S*-2-amino-3-[4′-*N*,*N*,-bis(2-chloroethyl)amino]phenyl propionic acid *N*-oxide dihydrochloride) to decrease HIF1α expression [32,33,34]. Firstly, we corroborated that the IH model increases mRNA expression of HIF1α and mTOR (Appendix A). Once the model was verified, our data showed that IH increases SMAD4 mRNA expression with respect to normoxic conditions, whereas SMAD4 expression was downregulated in PX478-treated cells (Figure 2C).

Together, these data show the effect of hypoxaemia (HIF1α and mTOR) on SMAD4 expression. Interestingly, we have also found that plasma Smad4 levels in OSA patients are related to the nocturnal recording time with oxyhaemoglobin saturation below 90% (CT90) (*r* = 0.290, *p* = 0.038, Figure 2D), a clinical parameter of hypoxaemia severity in OSA patients. 

### 3.4. The Activity of Circadian Rhythm Genes is Associated with SMAD4 Expression

To determine the circadian rhythm effect on the SMAD4 expression, we firstly analysed the main circadian player expression in OSA and HV samples, then, we analysed the correlation between the circadian genes and SMAD4 expression. Our data revealed a significant increased expression of BMAL1, NPAS2, CRY1, CRY2, PER1 and PER2 in OSA patients compared to HV subjects. Furthermore, in OSA patients, SMAD4 mRNA expression was significantly related to expression of BMAL1 (r = 0.3492, *p* = 0.0272), CLOCK (r = 0.373, *p* = 0.0164) and NPAS2 (r = 0.4108, *p* = 0.0085) (Figure 3A–C). In addition, a relationship was found between mRNA expression of SMAD4 and that of CRY1 and CRY2 (*r* = 0.4015, *p* = 0.0102, *r* = 0.3679, *p* = 0.0195, respectively). Furthermore, we found a significant correlation between SMAD4 expression and PER1 and PER2 mRNA expression (*r* = 0.4658, *p* = 0.0025 and *r* = 0.4989, *p* = 0.0011, respectively) (Figure 3D–G). Overall, we showed that circadian rhythm gene expression is also related to SMAD4 mRNA expression.

### 3.5. Association between Hypoxia and Circadian Genes 

In order to explore the possible association between hypoxaemia and circadian rhythm in OSA patients, we determined the association between HIF1α and circadian genes. Our data showed a significant positive correlation between HIF1α expression and that of CLOCK, PER1 and CRY1 (Figure 4). Interesting, the relationship between HIF1α and PER1 expressions suggested a possible synergic effect or link between OSA-induced hypoxaemia and circadian rhythm disorder. We have also explored the correlations between SMAD4 and BMAL1, NPAS2, PER2 and CRY2, but our data showed no statistical significance.

In addition, to support the hypothesis that SMAD4 expression could be elicited both by circadian genes and HIF1α, we performed a transcription factor binding site (TFBS) analysis of SMAD4’s best-supported promoter. Using the TRANSFAC^®^ database, we found five E-box and four hypoxia response elements (HRE), the positions of which are shown (Figure 5), and specified details such as position, sequences, matrix score and core similarity (Table 2).

### 3.6. Smad4 Expression is Associated with a Higher Risk of OSA Cardiometabolic Morbidity

Circulating Smad4 protein levels in plasma were higher in OSA patients with hypertension than in patients with normal blood pressure (625.6 ± 246.9 vs. 426.1 ± 225.0 pg/mL, *p* = 0.010) (Figure 6). Therefore, high plasma Smad4 levels were associated with the presence of hypertension (odds ratio [OR] per 50 pg/mL Smad4 increase 1.192, 95% confidence interval [CI] 1.031 to 1.378, *p* = 0.018), although the odds ratio adjusted for sex, age and body mass index (BMI) did not reach the threshold of statistical significance (adjusted OR 1.255, 95% CI 0.988 to 1.594, *p* = 0.063).

OSA patients with dyslipidaemia also showed higher plasma Smad4 levels than OSA patients with a normal lipid profile (634.5 ± 226.7 vs. 398.1 ± 238.7 pg/mL, *p* = 0.002) (Figure 6), identifying that each increase of 50 pg/mL in plasma levels of Smad4 raises the probability of concurrent dyslipidaemia, both in the crude analysis (OR 1.244, 95% CI 1.061 to 1.459, *p* = 0.007) and adjusted for age, sex and BMI (adjusted OR 1.318, 95% CI 1.047 to 1.659, *p* = 0.019).

Finally, type 2 diabetic OSA patients also have higher plasma Smad4 levels than OSA patients without diabetes (702.2 ± 192.4 vs. 460.7 ± 248.1 pg/mL, *p* = 0.002) (Figure 6). Thus, increased Smad4 levels were also a risk factor for a clinical diagnosis of diabetes (OR 1.301, 95% CI 1.071 to 1.580, *p* = 0.008), even after adjusting for age, sex, and BMI (adjusted OR 1.344, 95% CI 1.037 to 1.741, *p* = 0.025).

## 4. Discussion

The transforming growth factor (TGFβ) family of growth factors regulates several cellular responses related to the development and homeostasis of most human tissues. Smad4 is the last element of the SMAD complex which plays an important role as a regulator of TGFβ expression. As OSA is largely associated with metabolic, inflammatory comorbidities, we explored a possible relationship between SMAD4 overexpression and hypoxaemia or circadian rhythm parameters in these patients. We related SMAD4 expression with both hypoxaemia and circadian rhythm oscillation suggesting an explanation for the higher plasma Smad4 levels. In addition, we found that higher plasma levels of Smad4 are associated with the presence of several inflammatory or metabolic comorbidities in these patients. 

Previous studies have confirmed that both HIF1α and circadian genes induce TGFβ expression [6,24,25,26]. Interestingly, our results indicate that HIF1α and circadian genes induce SMAD4 expression. Therefore, this might indicate that both intermittent hypoxia and circadian rhythm disturbances induced by OSA might contribute to TGFß–Smad pathway activity. Secondly, our data suggest a synergic effect between circadian rhythm deregulation and hypoxaemia on SMAD4 levels in OSA patients. This potential interaction is supported by other studies proposing that there is a crosstalk between hypoxia response and regulation of circadian rhythm, demonstrating that both PER1 and BMAL1 interact with HIF1α, driving the transcription of common target genes [35,36,37]. Interestingly, HIF1α belongs to the same protein family as the core circadian proteins, the PAS domain superfamily of signal sensors for oxygen, light and metabolism [38,39]. PAS domains in PER proteins from circadian rhythm were found. The PAS domains shared between PER and HIF1a suggest their functions may be similar [39]. Moreover, there is solid evidence from in vitro assays to support the crosslink between circadian and HIF pathways [40]. ChIP-seq studies also revealed an E-box in the promoter for the HIF1α gene that was transcriptionally driven by binding of the BMAL1-CLOCK complex [41]. Interestingly, in an animal model study in mice subjected to hypoxia reported an increase in the levels of PER1 mRNA and CLOCK [35]. In line with this, our data showed a relationship between HIF1α mRNA and PER1, CLOCK and CRY1. Together, these data, gathered via various approaches, suggest that hypoxaemia or circadian genes, or a combination of these, contribute to increased SMAD4 expression in OSA patients. 

The circadian variation showed different functional effects on metabolic disorders and cardiovascular disease [42]. Recent studies have highlighted the relevance of circadian clock genes on the progression of several heart diseases and metabolic disorders. Disruption of the molecular clock causes atherosclerosis, insulin resistance, dampening of blood pressure rhythm, and a reduced production of vasoactive hormones and neurotransmitters [43]. Interestingly, in a BMAL1-knockout mice model, it has been demonstrated that blood pressure decreases by approximately 7 mmHg, that physiological heart rate and blood pressure rhythms are lost [44], and that dilated cardiomyopathy develops [45]. In addition, a missense mutation in the NPAS2 gene was found to be protective against hypertension [46]. Furthermore, the kidneys of hypertensive patients have been shown to have an increase in the expression of PER1 mRNA levels compared with normotensive subjects [47]. In line with this, some studies have linked circadian gene single nucleotide polymorphisms (SNPs) haplotypes with hypertension and type 2 diabetes [48]. Collectively, these data suggest the circadian clock to have a role in blood pressure regulation. However, the mechanisms underlying circadian gene involvement in blood pressure regulation remain undetermined. In this context, Xie and colleagues showed a mechanism by which BMAL1 regulates vasoconstriction via Rho-associated kinase 2 (ROCK2) in a manner dependent on the time of day [44]. Interestingly, ROCK2 transcription has been shown to be directly regulated by Smad4 [49]. This evidence suggests the direct role of circadian variation on SMAD4 expression. Our data support this hypothesis, because we found various potential E-BOX and HRE binding sites within SMAD4 promoter. Collectively, these data suggest that upregulated circadian genes activate the transcription of SMAD4. This assumption could explain the observed positive correlation between mRNA expression of SMAD4 and the circadian repressors genes PER/CRY, since both harbour E-box motifs and, thereby, their expression is activated by the same transcription factors. However, further analyses are needed in order to confirm these notions.

Furthermore, circadian rhythm has been also associated with obesity and insulin resistance through BMAL in an animal model [50]. Al-Sarraf and colleagues demonstrated that CRY2 is elevated in serum from patients with metabolic syndrome. The authors also reported a positive correlation between CRY2 and blood pressure, and waist circumference and plasma low-density lipoprotein cholesterol levels [51].

The growing interest in the interaction between IH and metabolic dysfunction is collected in a recent review [52]. In this regard, intermittent hypoxia modulates the pancreatic β cells [53], upregulates the hepatokines resulting the insulin resistance [54], exacerbates metabolic effects on diet-induced obesity [55] and the upregulation the adipokines [56]. Moreover, the intermittent hypoxia increased the systemic oxidative stress, endothelial dysfunction and rennin-angiotensin system activation augmenting the hypertension progress in animal model [57]. All this evidence in combination with our data contributes to understand the intermittent hypoxia mechanism underlying the OSA metabolic complications.

Moreover, in a prospective study, patients with type 2 diabetes exhibited high levels of TGFβ in plasma [9]. SMAD pathway is also highly related to the progression of the diabetes [58]. In line with this, TGFβ/SMAD3 signalling regulates insulin transcription in pancreatic islet β-cells, whereas SMAD3 deficiency in mice protects against insulin resistance and type 2 diabetes during high-fat-diet-induced obesity [59,60]. Furthermore, TGFβ/SMAD signalling in glucose-induced cell hypertrophy has been associated with insulin resistance and diabetes [61]. All these data, together with the TGFβ canonical pathway by SMAD complex, support that SMAD4 might play an important role in the development of some OSA complications such as diabetes.

In this study, we focused on SMAD4 expression due to the significant role of SMAD complex activity as a common mediator. The other Smad protein categories are related to receptor regulation (SMAD1, SMAD2, SMAD3, SMAD5 and SMAD8/9) or receptor inhibition (SMAD6 and SMAD7), while SMAD4 contributes to the overall regulation of both types and, therefore, better reflects the functionality of the SMAD cascade. Furthermore, the interaction between TGFβ/SMAD4 pathway and other well established pathways such as MAPK (mitogen-activated protein kinase), PI3K/AKT (phosphatidylinositol-3 kinase/AKT) and WNT/β-catenin pathways [62] support the importance of SMAD4 in OSA complications.

Our study has several limitations. First, it is a single-centre study with a limited sample size, although sufficient to detect the clinically relevant differences shown. Second, the cross-sectional design only allows evaluating relationships or potential associations. To assess causality a longitudinal study, with long term follow-up of the patients, would be necessary. However, this study design is not feasible from an ethical point of view, because we decided not to keep patients with severe OSA without treatment for an extended period of time. Third, the OSA patients were diagnosed using a validated respiratory polygraphy. Although polysomnography is the gold standard for OSA diagnosis, polygraphy is universally accepted and the most used technique in routine clinical practice, so we consider that our patients could be more representative and their characterization is adequate. Fourth, in this study we have only include severe OSA patients. Fifth, our study does not provide any information on the effect of OSA treatment on TGFβ/SMAD4 pathway activity or its effect on the progression of comorbidities.

## 5. Conclusions

In conclusion, our study shows that patients with severe OSA present an overexpression of SMAD4, induced both by intermittent hypoxaemia and circadian rhythm deregulation secondary to sleep fragmentation, reflecting greater activity of the TGFβ/SMAD pathway, and this being a risk factor for the presence of some comorbidities that involve TGFβ in their pathogenesis (Figure 7).

## Figures and Tables

**Figure 1 jcm-09-02378-f001:**
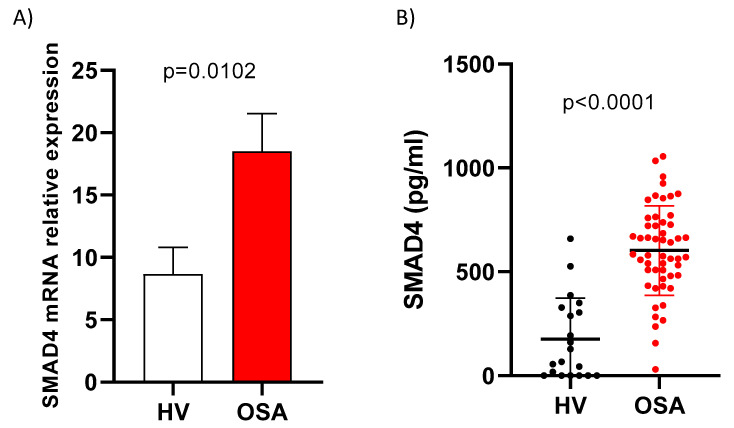
SMAD4 is upregulated in OSA patients. (**A**) The SMAD4 mRNA expression analysis by qPCR in monocytes from HV (*n* = 18) and severe OSA (*n* = 40). Comparison of mRNA expression levels between groups was performed by unpaired t-test with Welch’s test correction. Means ± SEM are shown. (**B**) Plasma from OSA patients and HV were collected to evaluate protein concentration. The Smad4 protein was quantified using ELISA technology (OSA *n* = 52 and HV *n* = 26). Comparison of protein levels between groups was performed by unpaired *t*-test. Means ± SEM are shown. *p*-values for OSA patients versus healthy volunteers are shown.

**Figure 2 jcm-09-02378-f002:**
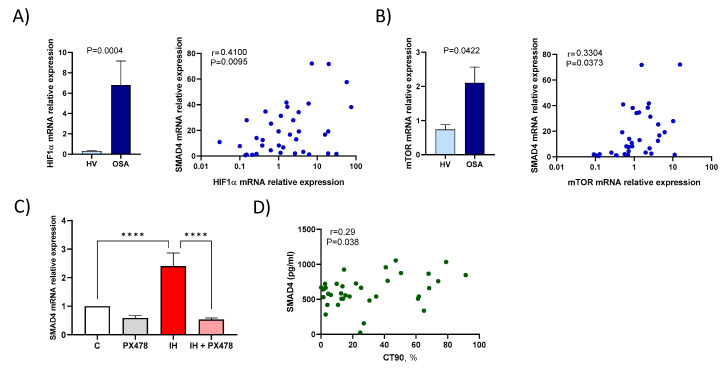
Hypoxaemia is related to SMAD4 expression in OSA patients. Relative expression of mRNA in monocytes from OSA patients (OSA) (*n* = 40) and healthy volunteers (HV) (*n* = 18). Correlation between relative expression of mRNA of HIF1α (**A**) or mTOR (**B**) with that of SMAD4 in monocytes from OSA patients (*n* = 40). OSA patients were randomly selected. Means ± SEM, Spearman’s correlation coefficients (r) and *p*-values are shown (**C**) SMAD4 mRNA expression analysis by qPCR in monocytes from HV (*n* = 7) treated with a specific inhibitor of HIF1α (PX-478 15 μM) and/or subjected to IH conditions for 16 h. Paired control samples were only incubated under normoxic conditions. Comparisons between groups were performed by two-way ANOVA. Means ± SEM are shown. **** *p* < 0.0001 cells are shown. (**D**) Relationship between nocturnal oxyhaemoglobin saturation lower than 90% (CT90) and plasma levels of Smad4 in OSA patients (*n* = 35). Spearman’s correlation coefficient and *p*-value are shown.

**Figure 3 jcm-09-02378-f003:**
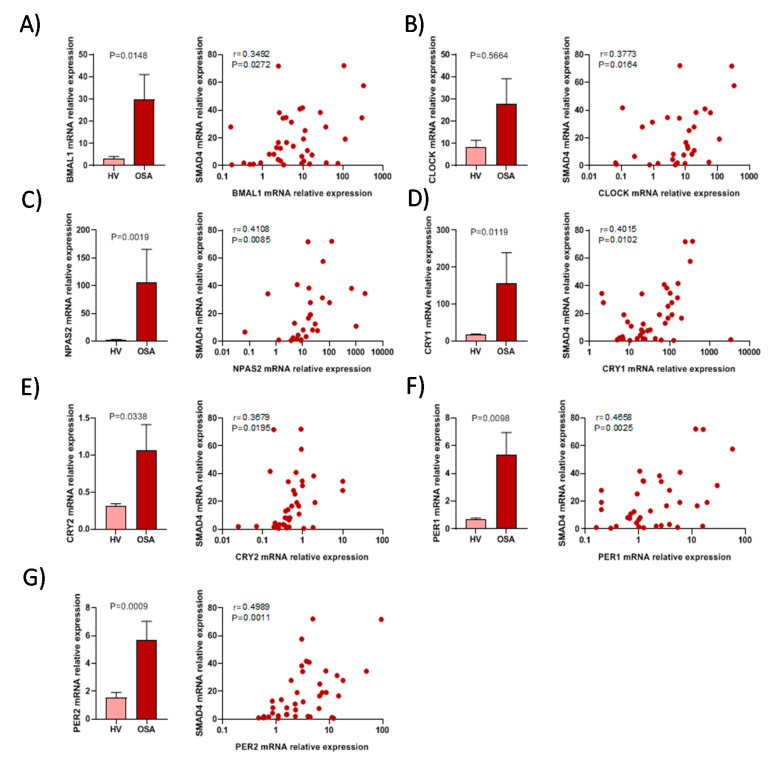
Circadian rhythm variation is related to SMAD4 expression in OSA patients. Relative mRNA expression in monocytes from OSA patients (OSA) (*n* = 40) and healthy volunteers (HV) (*n* = 18). Correlation between relative expression of circadian genes mRNA (BMAL1 (**A**), CLOCK (**B**), NPAS2 (**C**), CRY1 (**D**), CRY2 (**E**), PER1 (**F**) and PER2 (**G**)) and of SMAD4 mRNA in monocytes from OSA patients (*n* = 40). Comparisons between groups were performed by an unpaired t test with Welch’s correction. Means ± SEM are shown. Spearman’s correlation coefficients (r) and *p*-values are shown.

**Figure 4 jcm-09-02378-f004:**
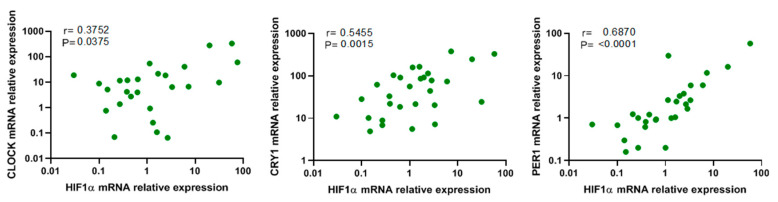
Circadian rhythm variation is related to HIF1α expression. Correlation between HIF1α mRNA expression and CLOCK (*n* = 40 **left** panel), CRY1 (*n* = 40, **middle** panel) and PER1 (*n* = 40, **right** panel) mRNA expression in monocytes from OSA patients. The OSA patients were randomly selected. Spearman’s correlation coefficients (r) and *p*-values are shown.

**Figure 5 jcm-09-02378-f005:**
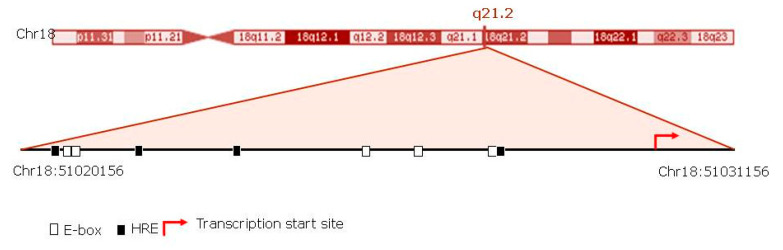
Schematic representation of the in silico analysis of the potential E-BOX and HRE binding sites, in the SMAD4 promoter sequence (Ensembl reference sequence ENS), based on the consensus sequence. Location of these E-BOX and HREs are shown. Data obtained from TRANSFAC^®^ database.

**Figure 6 jcm-09-02378-f006:**
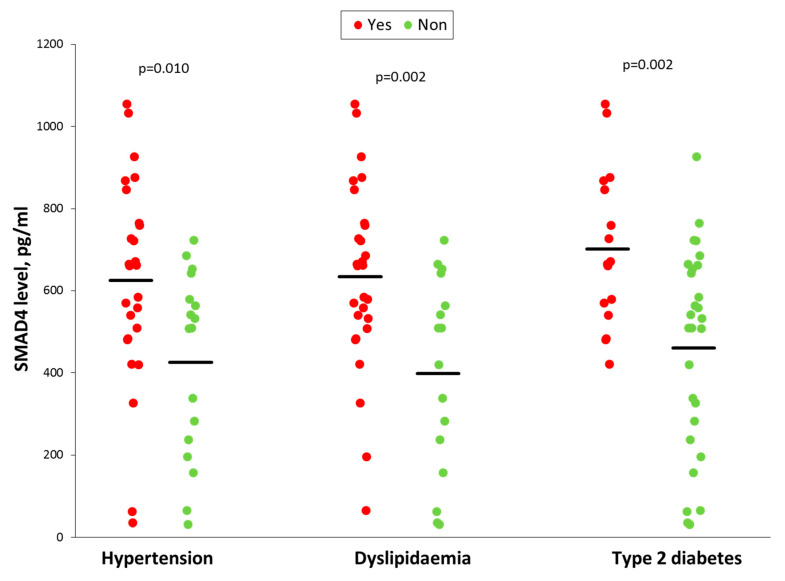
Comparison of plasma levels of Smad4 in OSA patients based on the presence or absence of hypertension, dyslipidaemia, or type 2 diabetes. Individual values are shown. Horizontal lines represent mean values. Comparisons were performed by t-Student test.

**Figure 7 jcm-09-02378-f007:**
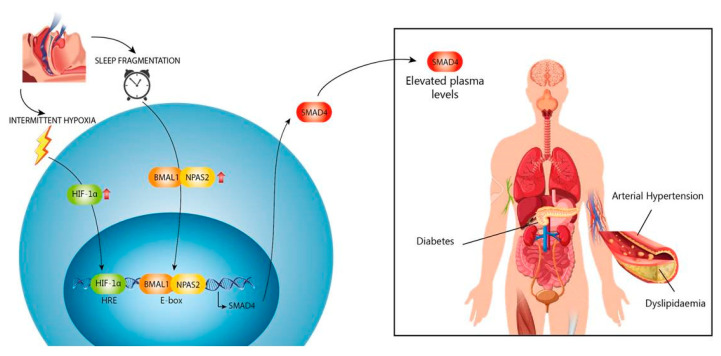
HIFα and mTOR are increased by intermittent hypoxia, and in combination with circadian rhythm deregulation, increase SMAD4 expression. High plasma levels of Smad4 are related to several cardiometabolic complications of OSA.

**Table 1 jcm-09-02378-t001:** General Characteristics of the Study Subjects.

Variable	Patients with Severe OSA (*n* = 52)	Healthy Volunteers (*n* = 26)	*p*-Value
Male Sex, *n* (%)	34 (65)	15 (58)	0.462
Age, Years	61 ± 12	59 ± 11	0.631
Body Mass Index, kg/m^2^	32.4 ± 5.4	29.3 ± 3.2	0.105
Smoking Habit			
Current Smoker	14 (28)	7 (27)	0.100
Former Smoker	18 (35)	8 (31)	
Never Smoker	20 (37)	11 (42)	
Epworth Sleepiness Scale	8.5 ± 4.2	6.0 ± 2.8	<0.001
AHI, Events/h	51.5 ± 16.8	2.8 ± 1.2	<0.001
Oxygen Desaturation Index, Events/h	45.9 ± 20.4	4.2 ± 4.5	<0.001
Time Recorded with SaO_2_ <90%, %	33.9 ± 29.4	4.4 ± 3.1	<0.001
Mean Nocturnal SaO_2_, %	91 ± 3	93 ± 2	0.037
Low Nocturnal SaO_2_, %	75 ± 8	85 ± 5	0.001
Comorbidities			
Hypertension, *n* (%)	28 (54)	0	<0.001
Dyslipidaemia, *n* (%)	29 (56)	0	<0.001
Type 2 diabetes, *n* (%)	16 (31)	0	<0.001
Systolic BP, mmHg	131 ± 19	125 ± 9	0.009
Diastolic BP, mmHg	80 ± 10	76 ± 8	0.043
Cholesterol, mg/dL	217 ± 51	186 ± 45	<0.001
HDL-cholesterol, mg/dL	46 ± 15	49 ± 14	0.028
LDL-cholesterol, mg/dL	148 ± 51	118 ± 40	<0.001
Triglycerides, mg/dL	153 ± 77	137 ± 61	0.037
Fasting glycaemia, mg/dL	119 ± 38	108 ± 26	0.089
Haemoglobin A1c	6.1 ± 1.1	5.4 ± 0.8	0.011

Data are expressed as mean ± SD or number (percentage). Comparisons between groups were performed by t-Student test or the chi-squared test. Abbreviations: AHI = apnoea–hypopnoea index; SaO_2_ = oxyhaemoglobin saturation; BP = blood pressure; HDL = high-density lipoproteins; LDL = low density lipoproteins.

**Table 2 jcm-09-02378-t002:** Position, matrix score, core score and sequence from the EBOX and HRE motifs.

Motif	Position	Matrix Score	Core Score	Sequence
EBOX	51020730 (−)	0.930	0.932	ACACATGG
EBOX	51020732 (−)	1.000	0.987	ACACATGG
EBOX	51025411 (+)	1.000	0.992	ACATGTGT
EBOX	51026204 (+)	1.000	0.983	GCATGTGT
EBOX	51027428 (−)	0.932	0.920	CCACCTGC
HRE	51027681 (−)	1.000	0.989	CACGTCC
HRE	51021767 (+)	1.000	0.956	AACGTGG
HRE	51020718 (+)	1.000	1.000	CACGTAC
HRE	51023404 (−)	1.000	1.000	TACGTGA

Data were obtained from the TRANSFAC^®^ database.

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
