# Peer review of "SMAD4 Overexpression in Patients with Sleep Apnoea May Be Associated with Cardiometabolic Comorbidities"

_jcm, 2020, doi:10.3390/jcm9082378_

Round 1

Reviewer 1 Report

I have several questions for the authors:

1) SMAD4 is generally considered to be an intracellular protein. It is also generally considered as the "co-SMAD" and partner protein to the other SMAD family members that are directly responding to extracellular signals, rather than a signalling mediator and transcriptional regulator itself. How do the authors justify their decision to look for (and indeed find it) in plasma? Figure 7 shows a speculative mechanism implying that SMAD4 is exported from cells - how and why would this happen?

2) PBMCs are a relatively easily accessible cell type, and logically would be highly exposed to hypoxaemia, but why would they be particularly affected by circadian dysregulation? There is no justification of the choice of this cell type. 

3) In my opinion the jump from OSA and intermittent hypoxaemia to circadian dysregulation is tenuous. There is no mention or description of the circadian status of the donors at the time of collection (e.g. salivary cortisol collections or similar).

4) In figure 2B, the legend states that P values were from t tests. For two comparisons with the same control comparator, please justify why this was chosen rather than a repeated measures ANOVA? In other panels of figure 2, and in figure 3, the 

5) In figure 3, the distribution of SMAD4 expression seems different between the different panels shown. In several of the panels there seem to be a small number of samples that are driving the observed relationships. Were the same 44 samples used for all comparisons?

6) The potential transcriptional regulatory binding sites listed in table 2 appear to have been obtained from a simple database search. Do the authors have any indication that these sites play any functional role in this context? 

7) I am a little confused about the apparently interchangeable use of SMAD4 and Smad4 in the text (particularly page 8). Please ensure this is consistent.

8) The odds ratios presented for the logistic regression analyses for hypertension, dyslipidaemia and type 2 diabetes are very small (close to 1) as I presume the SMAD4 level has been used in pg/ml units, which are distributed over a very wide range. It might be more informative to present these as odds ratios per 50pg/ml increase, for example. Also in this section, it is important not to conflate odds with risk, as statistically these are not the same.

Reviewer 2 Report

This manuscript aims to 1) analyze the relationship between indices of hypoxaemia or circadian rhythm and SMAD4 expression in OSA patients; 2) explore the association between SMAD4 expression and cardiometabolic comorbidities in OSA patients. Authors concluded that their study shows that patients with severe OSA present an overexpression of SMAD4, induced both by intermittent hypoxemia and circadian rhythm deregulation secondary to sleep fragmentation, reflecting greater activity of the TGFβ/SMAD pathway, and this being a risk factor for the presence of some comorbidities that involve TGFβ in their pathogenesis.

Although such results could be interesting for the understanding of OSA pathophysiology and related complications, the study needs major revisions. Most of the conclusions are overstated and based only on correlations. Several data, such as complete flow chart of the study and more precise characterization of subjects included in the study, are lacking.

Major

  1. Introduction would benefit from precisions, with a paragraph introducing the potential link between sleep fragmentation and/or circadian rhythm with TGFb-Smad pathway. In the same manner, there is a well-documented literature concerning the potential cross-talk between HIF-1 and TGF-beta pathways that should be added in the introduction. Indeed, it is well described that HIF-1 could activate TGFb signaling and that conversely TGFb could activate HIF-1.
  2. The flow chart of the study must be included in the methods section. In line with this, authors have to verify all the general parameters as there are discrepancies between text and table. In the text, authors stated that the included thirty healthy volunteers, whereas there are only 26 in table 1. As authors did subgroups analysis (in Figure 6), they must provide in table 1 data justifying hypertension (blood pressure), dyslipidemia (circulating lipid parameters (LDL-c, HDL-c)) and type 2 diabetes (glycemia, insulinemia and HOMA-IR).
  3. Why number of subjects are so different between experiments and figures? Sometimes n=37, another time n=18 or 34 or even 44? This should be explained using a complete flow chart or separate additional figures/tables summarizing all the genes analyzed or ELISA performed, on which number of subjects?
  4. In the methods section, how is mRNA results expressed. In the figures, it is indicated that results are expressed as relative expression. Compared to what? A control group? A reference gene? Both? By which methods (i.e. 2-ΔΔCT method)?
  5. Cell exposure to intermittent hypoxia refers to a precise description in previous papers [22, 23]. However, it seems that the model changed between these studies. Indeed, ref 23 described cycles with 5 min hypoxia 3% and 10 minutes normoxia, which were achieved through changes of medium, whereas in the present study, cycles durations are 2 min hypoxia 1% followed by 10 min normoxia 20%. This must be clarified and explained precisely. In addition, please indicate the duration to reach both hypoxic or normoxic plateaus, and also precise the total IH exposure duration (number of cycles per day, number of days).
  6. Results are presented only through correlations. The manuscript would benefit to also present gene expressions (HIF1, mTOR, all the genes related to circadian rhythm) through bar graphs, comparing OSA patients to healthy volunteers.
  7. Authors assimilate hypoxemia to HIF1 expression. This is an overinterpretation that should be stated with more caution. Also, could the authors show a bar graph of HIF1 expression in the OSA patients and HV ? As authors present ODI and saturation of their subject, did these parameters (related to hypoxemia) correlate with Smad4 expression?
  8. As mentioned above, figure 6 must be accompanied by data in table 1, showing the 3 subgroups (hypertension, dyslipidemia, type 2 diabetes). In addition, this should be added in the method section (i.e. how are defined these comorbidities?)
  9. Results are overinterpreted in both discussion and conclusions. For example, results of the present study did not allow to conclude that: “We identified the mechanism by which SMAD4 expression is increased in patients with OSA” as the authors only correlated some factors, but not afforded mechanistic demonstration. To conclude on such mechanisms involvement, it would have been interesting to design a more robust experiment using the cellular model of intermittent hypoxia, targeting both HIF1 and CLOCK pathways (pharmacology, siRNA…) and analyzing the impact of inhibition/deletion on TGFb-Smad4.
    In the same manner, it is not possible to conclude on “a synergic effect between circadian rhythm deregulation and hypoxaemia, increasing SMAD4 levels in OSA patients.” The authors should be very careful with their data interpretations.

Minor:

  1. Abstract should be written following the classical structuration: background, methods, results and conclusion
  2. For all the graph, it should be indicated how are expressed results (i.e. mean ± sem or mean ± SD or median…). Also, this should be uniformized between all the graphs. Sometimes SEM are used, sometimes are SD.
  3. In the following sentence (page 2, line 59): “intermittent hypoxia (IH) triggers HIF1α expression [6], which has been associated with TGFβ overexpression [21].” Please verify both references, as they did not seems to match with the statement.

Reviewer 3 Report

Diaz-Garcia et al has conducted a case-controlled study to examine SMAD4 expression in individuals with severe sleep apnea compared to controls. Additionally they have then explored the association between SMAD4 expression and molecular markers of intermittent hypoxia and circadian rhythm regulation as well as clinical conditions such as hypertension, diabetes and dyslipidaemia.

Comments

  1. Only individuals with severe OSA have been investigated. Why was mild-moderate OSA excluded? Would there be a dose-dependent response? How this may affect generalisability of these results should be discussed.
  2. Again reasoning for the limit of 35 years and up? Was there a maximum age limit?
  3. How were the healthy controls recruited? Were they from the sleep laboratory or the community? How were they matched for age, BMI etc- was this on a one-to-one basis or just on means levels? An AHI is reported for controls so were they people just coming through the sleep lab or were they recruited for the community and then OSA screened out using a PSG.
  4. The statistical section should be clarified. As this is a case-controlled study it was not clear why paired ttests were used until I reached the results section and saw it was for an exploratory analysis. Also making clear when Pearson (stated in stats section) and spearmen’s (stated in figure legends) were used. Also see point 6.
  5. The p values shown on Figure 1 seem to be different to that in the text. Also in the figure legend these don’t seem to match either the figure or the text?
  6. All correlation figures state that Spearman’s was used however an r value is reported rather than a rho coefficient. Please confirm whether Spearmans or Pearson’s are shown and correct if Spearman’s. If Spearman then a linear regression line should not be drawn. A number of correlations do seem to be possibly reliant on outliers. Was this investigated?
  7. Was hypertension, dyslipidaemia and diabetes defined by self-reported medical history? Were any blood tests for glucose or lipids conducted or blood pressure assessed at the PSG visit?
  8. Limitations of the study design (including sample size, how controls are defined, how medical conditions are defined, cross-sectional design etc) need to be highlighted in the discussion and how this may have affected the results and implications. Additionally clinical applications or relevance could be highlighted further.

Round 2

Reviewer 2 Report

Many mistakes are still present and some new questions raised from this new version.

Some errors remains in the references. In example, authors claim that they removed ref 22. It is the case in the method section, however, ref 22 is still used to describe IH model in cell culture in the result section. Also, some reference are not formatted appropriately in the bibliography.

Requested data have been included, however, description of results have not been modified appropriately. In example, HIF1 and mTOR mRNA expressions in figure 2 are presented before correlation, whereas they appear at the end of the result description in the result section.

Flow chart affords some explanations in the study design, however, some discrepancies between the 2 versions of the manuscript appears. How is it possible that in the first version of the manuscript, mRNA expressions were reported in n=44, whereas flow chart reports that monocyte mRNA expression was analyzed on  n=40 OSA patients? In the same way, all the correlations showed in figures 2 and 3 have been changed between the 2 versions. Why?

Concerning mRNA analysis, the gene expression quantification is still not clear. Along the results, it seems that results have not all been analyzed through the same methods. Some graphs show  normalization of the HV group to 1, with no error bars (figure 2C and supplementary figure 2), whereas in other graph HV group is not normalized to 1 and exhibit error bars. It seems improbable that no variability exist in this HV group. But if so, how are the statistics realized?

To conclude on results from figure 2B, the group IH+PX478 is necessary to show that IH-induced SMAD4 expression is HIF1-dependant, and such results have to be analyzed through a two-way ANOVA (one factor=hypoxic stimulus, other factor=treatment with PX478). As Smad4 levels (mRNA and proteins) are quite low in HV subjects, the pertinence to investigate the effect of HIF1 inhibitor in this group is questionable.

In the review, authors present a new figure that show a correlation between HIF1 mRNA expression and both AHI and ODI. This figure is named Figure 2, however, do not correspond to figure 2 in the revised manuscript.

The very slight modifications made in the discussion to modulate the message are not very convincing.

Overall, the responses to reviewer have been uploaded very rapidly after review submission and it seems that it has been down in a rush. Authors should return very carefully to all their data, their description and interpretation.

Reviewer 3 Report

I thank the authors for addressing the queries. I feel the responses are adequate. I would however suggest that the authors add in the methodology regarding the selection of control participants. This is an important feature of the STROBE guidelines for reporting of case-control studies (see Strobe checklist). 
